# *Odontites linkii* subsp. *cyprius* Ethanolic Extract Indicated In Vitro Anti-*Acanthamoeba* Effect

**DOI:** 10.3390/microorganisms12112303

**Published:** 2024-11-13

**Authors:** Chad Schou, Zeynep Kolören, Jandirk Sendker, Yiannis Sarigiannis, Aleksandar Jovanovic, Panagiotis Karanis

**Affiliations:** 1Department of Basic and Clinical Sciences, University of Nicosia Medical School, CY-1700 Nicosia, Cyprus; schou.c@unic.ac.cy (C.S.); jovanovic.a@unic.ac.cy (A.J.); 2Department of Molecular Biology and Genetics, Faculty of Arts and Sciences, Ordu University, 5200 Altınordu, Ordu, Turkey; zeynep.koloren@gmail.com; 3Institute of Pharmaceutical Biology and Phytochemistry (IPBP), University of Münster, PharmaCampus, Corrensstraße 48, 48149 Münster, Germany; jandirk.sendker@uni-muenster.de; 4Department of Health Sciences, School of Life and Health Sciences, University of Nicosia, CY-2417 Nicosia, Cyprus; sarigiannis.i@unic.ac.cy; 5Bioactive Molecules Research Center, School of Life & Health Sciences, University of Nicosia, CY-2417 Nicosia, Cyprus; 6Center for Neuroscience and Integrative Brain Research (CENIBRE), University of Nicosia Medical School, CY-1700 Nicosia, Cyprus; 7Faculty of Medicine and University Hospital, University of Cologne, 50937 Cologne, Germany

**Keywords:** anti-*Acanthamoeba* effect, ethanolic leaf extracts, *Odontites linkii* subsp. *cyprius*, in vitro

## Abstract

This study aims to investigate three endemic ethanolic leaf extracts from Cyprus for anti-*Acanthamoeba* activities: *Odontites linkii* subsp. *cyprius* (Boiss.) Bolliger, *Ptilostemon chamaepeuce* subsp. *cyprius* (Greuter) Chrtek & B. Slavík, and Quercus *alnifolia* Poech. Screening for radical scavenging activity, total phenolic content (TPC), and total flavonoid content (TFC) were performed by the 2,2-diphenyl-1-picrylhydrazyl (DPPH) and 2,2′-azino-bis(3-ethylbenzothiazoline-6-sulfonic acid (ABST) methods, Folin–Ciocalteu method, and aluminum chloride method, respectively. An antibacterial-susceptibility test (AST) was performed using a broth microdilution assay to estimate the minimum inhibitory concentration (MIC) using iodonitrotetrazolium chloride (INT). Trypan blue (0.5%) was used to assess in vitro anti-*Acanthamoeba* cell viability of the ethanolic leaf extracts after 24-, 48-, and 72-h exposure—screening of ethanolic leaf extracts with liquid chromatography–mass spectrometry (LC-MS) for known compounds with biological activity. The ethanolic leaf extract of *Odontites linkii* subsp. *cyprius* demonstrated the highest anti-*Acanthamoeba* activity, with an inhibitory concentration (IC_50_) of 7.02 mg/mL after 72 h. This extract also showed an in vitro minimum inhibitory concentration (MIC) of 0.625 mg/mL against *Enterococcus faecalis*, a common nosocomial pathogen. The LC-MS analysis revealed the presence of bioactive iridoid compounds in *O. linkii* subsp. *cyprius*, further highlighting its potential as a source for new drug compounds. The ethanolic extract of *O*. *linkii* subsp. *cyprius* demonstrated a dose-dependent and time-dependent anti-*Acanthamoeba* effect in vitro. This study is the first to report the presence of iridoid compounds and anti-*Acanthamoeba* activities in the ethanolic extract of *O. linkii* subsp. *cyprius.* These promising findings highlight the potential of plant extracts, particularly *O. linkii* subsp. *cyprius*, as a source for new drug compounds for *Acanthamoeba* infections.

## 1. Introduction

### Acanthamoeba Infection

Free-living amoebae (FLA) belong to the genera *Acanthamoeba*, *Balamuthia*, *Naegleria*, and *Sappinia* and can cause disease and death when they infect animals and humans. *Acanthamoeba* species (spp.) are opportunistic protists and are known to cause severe keratitis among contact lens wearers and can trigger cutaneous lesions, especially among immunocompromised individuals [1,2,3]. Another free-living amoeba, *Naegleria fowleri*, can produce an acute and typically lethal central nervous system (CNS) disease called primary amoebic meningoencephalitis (PAM) [4]. Infections from free-living amoebas are not considered neglected tropical diseases but can become a public health concern if environmental circumstances encourage reproductive conditions [5,6,7]. *Acanthamoeba* spp. infections are the most common cause of *Acanthamoeba* keratitis (AK). They can also cause granulomatous amoebic encephalitis (GAE) in humans with compromised immune systems, especially if the patient is suffering from a chronic debilitating illness, human immunodeficiency virus (HIV) infection, or an organ transplant [8,9]. 

Due to a high prevalence in the environment, *Acanthamoeba* spp. have been proposed to serve as a reservoir for some human and animal pathogenic bacteria. *Legionella pneumophila*, *Pseudomonas aeruginosa*, and certain strains of *Escherichia coli* and *Staphylococcus aureus* can infect and replicate within *Acanthamoeba* spp., which could lead to antibiotic resistance in these bacteria as they congregate within the amoebas [10]. In addition, *Acanthamoeba* spp. have been documented to uptake and transport oocysts of the protozoan parasite *Cryptosporidium parvum* [11]. 

*Acanthamoeba* spp. can tolerate harsh environmental conditions due to survival mechanisms. *Acanthamoeba* spp. have two forms, the trophozoite (metabolically active) and a dormant, stress-resistant cyst, which has allowed it to be isolated from chlorinated swimming pools, domestic tap water, soil, air, sewage, seawater, bottled water, hospitals, air-conditioning units, and contact lens cases [12,13,14,15]. Isolates have also been recovered from nasal cavities, skin, and intestines, as well as plants and other vertebrates [1]. 

Due to the lack of effective chemotherapies and often late diagnosis, the treatment of *Acanthamoeba* spp. infections is a real challenge. Acid-resistant proteins (outside wall) and cellulose (inner wall) are the major components of the *Acanthamoeba* cyst wall. The double-wall cyst stage allows this protist to survive harsh environments and available treatments, which leads to patient reinfection and becomes life-threatening [16]. Miltefosine has shown promise as a potential treatment for granulomatous amoebic encephalitis and *Acanthamoeba* keratitis that is resistant to standard therapy [17,18,19]. This alkylated phosphocholine was originally investigated as an antitumor agent, which has been approved as an oral antileishmanial treatment [20]. Side effects such as abdominal pain, diarrhea, nausea, and vomiting have been reported in the literature that may limit its use in patients. 

In this study, we report the potential of ethanolic leaf extracts from three endemic plants from Cyprus for their effectiveness against *A. castellanii* trophozoites in vitro. The extracts were also evaluated for their ability to scavenge free radicals and their total phenolic and flavonoid content. Additionally, we assessed their antibacterial properties using a broth microdilution method to determine the minimum inhibitory concentration (MIC) against six reference pathogenic bacteria. The toxicity of these extracts was tested on human hepatocellular carcinoma (HepG2) cells. Finally, the extracts were analyzed using liquid chromatography–mass spectrometry (LC-MS) to identify known compounds with biological activity.

## 2. Materials and Methods

### 2.1. Cyprus Phytogeography

Cyprus, the third largest island in the Mediterranean, boasts a unique biodiversity due to its two dominant mountain ranges and a central plain. The Kyrenia Mountain range, a narrow limestone ridge, and the larger Troodos Mountain range, primarily consisting of molten igneous rock, create a diverse environment in terms of altitude, temperature, rainfall, and soil composition. This unique environment has fostered a rich flora and fauna diversity on the island, with Cyprus sharing a similar phytogeographical relationship to Asia Minor and Syria.

The flora of Cyprus consists of around 1649 taxa, of which 146 are endemic to Cyprus [21] and are only found growing in specific habitats. Most of these endemic plants have not been characterized for antimicrobial potential. Investigating bioactive plant compounds from these endemic plants could lead to the development of better drugs to treat diseases and acquired infections. 

Unfortunately, limited pharmaceutical antimicrobial and antiparasitic choices exist to treat *Acanthamoeba* spp. infections. New plant-based compounds need to be investigated for improved treatments of *Acanthamoeba* spp. infections.

This study was conducted from June 2022 to July 2024. Plants were collected from Cyprus, and dried ethanolic extracts were prepared and shipped to the Department of Molecular Biology and Genetics, Faculty of Arts and Sciences at the University of Ordu, Turkey, for the anti-*Acanthamoeba* assay. Antibacterial-susceptibility test (broth microdilution assay), TPC and TFC assays, and DPPH and ABST radical scavenging assays were performed at the Department of Basic and Clinical Sciences, University of Nicosia Medical School, Cyprus. Liquid chromatography–mass spectrometry (LC-MS) analysis was performed at the Institute of Pharmaceutical Biology and Phytochemistry, University of Münster, PharmaCampus, Münster, Germany. 

### 2.2. Collection of Plant Materials and Preparation of Ethanolic Plant Extracts

Leaves were collected from Cyprus collection sites (Figure 1) in June 2022 and identified by Elefterios Hadjisterkotis (Environmental Office at the Cyprus Agriculture Research Institute). Voucher specimens (Table 1) were placed at the Department of Basic and Clinical Sciences at the University of Nicosia Medical School. The leaves were washed and air-dried in the shade for 3 weeks and ground to a fine powder with an electric mill. Afterwards, they were stored at 4 °C in plastic zipper bags. For extraction, 1 g of finely powdered plant leaves was combined with 20 mL absolute ethanol in a clean 50 mL plastic centrifuge tube and ultrasonicated at full power for 20 min at 40 °C. This extraction process was repeated once; both supernatants were combined and vacuum filtered with a Whatman No. 1 paper. This process was further repeated until 4–5 g of plant material had been extracted. Using a rotary evaporator, the extracts were then concentrated to dryness in a tared glass vial under reduced pressure at 40 °C. The dried extracts were stored at 4 °C until used for investigations. 

### 2.3. DPPH (2,2-Diphenyl-1-picrylhydrazyl) Radical Scavenging Activity

The DPPH assay used in this study was adapted from a method developed by Shimamura et al. (2014) [22]. Small volumes from dry extracts (0.2 mL) were combined in 0.1 M Tris-HCl (pH 7.4) solution and mixed with 1 mL of a 0.2 mM DPPH in ethanol. The absorbance at 517 nm of the radical scavenging activity was measured after a 30 min incubation at 25 °C in darkness with a Jasco V-730 spectrophotometer (Hachioji, Tokyo, Japan). Blank control consisted of 0.2 mL EtOH added to 0.8 mL of 0.1 M Tris-HCl (pH 7.4) buffer and 1 mL DPPH solution. Negative consisted of 1.2 mL EtOH added to 0.8 mL 0.1 M Tris-HCl (pH 7.4). Freshly prepared ascorbic acid and Trolox (vitamin E analog) were used as the antioxidant references. The percent radical scavenging activity (%RSA) was calculated with the following formula: %RSA = (Abs control − Abs sample)/(control) × 100. The percent inhibition at 50% concentration (IC_50_) was calculated for each plant extract and Trolox and ascorbic acid by plotting inhibition ratios (y) against sample concentrations (x) at all six points and then drawing a regression line (y = ax + b). The activity was expressed as IC_50_ (inhibitory concentration of each extract that scavenges 50% of DPPH radicals). IC_50_ = 50 = ax + b (mg/mL) when y = 50.

### 2.4. ABTS (2,2′-Azino-Bis(3-ethylbenzothiazoline-6-sulfonic Acid) Radical Scavenging Activity

The radical scavenging capacity of the plant extracts was also evaluated with the ABTS assay to compare it to the DPPH radical scavenging capacity of the extracts. The ability of the plant extracts to neutralize ABTS•+ radical cation was assayed using the procedure outlined by Re et al. (1999) [23]. In brief, 0.1 mL diluted samples/Trolox standards were added to 2 mL ABTS working solution, where the blank consisted of 0.1 mL EtOH added to 2 mL of ABTS working solution. Freshly prepared ascorbic acid was the positive control. The absorbances at 734 nm were recorded in triplicate against blank by a V-730 Jasco UV/Vis Spectrophotometer (Hachioji, Tokyo, Japan). The mean of three replicates was used for the calculations. Radical scavenging capacity (RSA%) = 100 − [absorbance of sample − absorbance of sample blank) × 100/(absorbance of control) − (absorbance of control blank)]. The IC_50_ values were calculated from the graph plotted as inhibition percentage against the concentration. 

### 2.5. Determination of Total Phenolic Content (TPC)

The total phenolic content of the ethanolic extracts was quantified spectrophotometrically by the Folin–Ciocalteu (F-C) method, as described according to Singleton et al. (1999) [24], where gallic acid served as a standard. In brief, 0.25 mL of diluted sample/standard was combined with 1.25 mL F-C reagent (1:10) in darkness for six minutes at room temperature. Afterwards, 1 mL of 7.5% sodium carbonate was added to each serial dilution and allowed to react for two hours in darkness at room temperature. Blank was prepared with absolute ethanol instead of sample/standard. At 760 nm, the UV absorbances of the serial dilutions against blank were recorded in triplicate by a V-730 Jasco UV/Vis Spectrophotometer (Hachioji, Tokyo, Japan). The results were expressed as mg gallic acid equivalents per gram of dry extract (mg GAE/g).

### 2.6. Total Flavonoid Content (TFC)

The total flavonoid content from the plant extracts was determined by the aluminum chloride colorimetric method according to Chang et al. (2002) [25], with slight modification. In brief, 0.5 mL of diluted sample/standard was combined with 1.5 mL absolute ethanol and mixed with 0.1 mL of a 10% aluminum chloride solution. Afterwards, 0.1 mL of a 1 M sodium acetate solution was added and vortexed. Distilled water was added to give a final volume of 5 mL, and the mixture was incubated at 25 °C in darkness for thirty minutes. Quercetin was used as a standard. Blank consisted of 2 mL absolute ethanol, 0.1 mL distilled water, 0.1 mL of 1 M sodium acetate, and 2.8 mL distilled water. The UV absorbances of the reaction mixtures were measured against blank at 415 nm in triplicate using a V-730 Jasco UV/Vis Spectrophotometer (Hachioji, Tokyo, Japan). The total flavonoid content of the extract was expressed as milligram quercetin equivalents per gram of dry extract (mg QE/g).

### 2.7. Antibacterial-Susceptibility Test (AST)—Broth Microdilution Assay

The broth microdilution assay was used to evaluate the antibacterial activity of the three endemic plants against six pathogenic ATCC reference bacteria. The ethanolic leaf extracts were meticulously screened for the minimal inhibitory concentration (MIC) according to the method outlined by Eloff (1998) [26] against the following ATCC^®^ reference pathogenic bacteria: Gram-positive—*Bacillus subtilis* (BS) (ATCC 6633), *Enterococcus faecalis* (EF) (ATCC 29212), *Staphylococcus aureus* (SA) (ATCC 6538), and *S. epidermis* (SE) (ATCC 12228) and Gram-negative—*Escherichia coli* (EC) (ATCC 25922) and *Pseudomonas aeruginosa* (PA) (ATCC 27853). 

A 100 µL volume of each plant extract was serial diluted with sterile saline in a sterile 96-well plate, and 100 µL of ATCC reference bacteria was added to yield a final plant extract concentration range of 2.5–0.02 mg/mL. The final assay inoculum 5 × 10^5^ cfu/mL was prepared by freshly diluting a 0.5 McFarland suspension of bacteria in sterile Mueller Hinton broth. Afterwards, the 96-well plate was incubated at 37 °C for 18 h. To visually determine the minimal inhibitory concentration (MIC), 40 µL of a 0.2 mg/mL p-iodonitrotetrazolium chloride (INT) solution was added to each well to evaluate the extent of antibiotic activity of each plant extract. The plates were incubated for one hour at 37 °C in darkness until INT was reduced to a pink color in the presence of metabolically active bacteria. Blank, background plant extracts, and solvent (0.5% ethanol, final) were accounted for in the assay. Gentamicin was used as the positive control at the same final dilutions as the plant extracts. 

### 2.8. Anti-Acanthamoeba Assay

#### 2.8.1. In Vitro Culture of Acanthamoeba 

A reference strain of *Acanthamoeba castellanii* (ATCC 30010) from the American Type Culture Collection was used in this study. *A. castellanii* strain was cultured on Ringer agar plates seeded with *E. coli* (bacteria) as a nutrient source. The plates were kept at 26 °C in the incubator, and three days later, they were checked under the microscope for *Acanthamoeba* trophozoites. When the trophozoites reached the stage of exponential growth (72 h), they were gently removed from the Ringer agar plates using a sterile cell scraper. They were washed twice with Ringer dilution and concentrated with a series of centrifugal processes in 1.5 mL sterile tubes according to the method described by Kolören et al. (2019) [27]. Trypan blue staining was used to determine the number of viable trophozoites by counting on a hemocytometer. The final assay concentration was 8 × 10^5^ trophozoites/mL. 

#### 2.8.2. Determination of the Anti-Acanthamoeba Activity In Vitro 

The % cell viability was used to screen the leaf extracts for anti-*Acanthamoeba* activity. The concentrated crude plant extracts were dissolved in 10% DMSO and distilled water to a final volume of 100 µL that was added to 100 µL volume of amoebae culture in the following final assay concentrations: 3.125, 6.25, 12.5, and 25 mg/mL. The *A. castellani* viability was checked with a microscope (×20) at 24-, 48-, and 72-h intervals using a Thoma hemocytometer chamber for the determination of the anti-amoebicidal activity of the plant extracts. *Acanthamoeba* % cell viability was determined by 0.5% trypan blue exclusion staining. All tests were repeated three times. The control group was a culture of amoebae in distilled water with DMSO and without extract. The mean results are given as % inhibition compared to control cells (considered as 100%). 

### 2.9. In Vitro Human Hepatocarcinoma Cell Line (HepG2) Resazurin Cell Viability Assay

A 100 µL volume of HepG2 cells (1 × 10^5^ cells/mL) was seeded in sterile 96-well plates with black walls and clear bottoms (Santa Cruz Biotechnology, Dallas, TX, USA) for 24 h. The overnight medium was replaced with 200 µL of fresh medium containing the diluted plant extracts in triplicate and allowed to incubate for 48 h at 37 °C and 5% CO_2_ atmosphere. Final assay plant extract concentrations were 0, 7.81, 15.63, 31.25, 62.5, 125, 250, and 500 µg/mL. Miltefosine was used as the positive control at final assay concentrations of 0, 15, 30, 60, and 120 µg/mL. 

To evaluate the plant extracts’ cell toxicity, resazurin was freshly prepared in 1× PBS at a final concentration of 20 µg/mL and passed through a 0.2 µm sterile PES syringe disc before adding 20 µL to each well. Plates were returned to the same CO_2_ incubator for another 4 h before fluorescence was measured at 550 nm excitation wavelength and 590 nm emission in a Jasco FP-850 spectrofluorometer plate reader (Hachioji, Tokyo, Japan). Each assay included blank, solvent, and plant extract backgrounds as quality controls. The experiment was repeated twice. The percentage of cytotoxicity compared to the untreated cells was determined with the equation: % Cell viability = control − treatment/control × 100. Results were expressed as CC_50_ values, as the concentration of compound that inhibits cell growth by 50% compared to control (no treatment). The CC_50_ values for cytotoxicity assay were calculated from a nonlinear regression analysis (curve fit) based on the sigmoid dose–response curve (variable) and run using GraphPad Prism 10 (GraphPad Software, San Diego, CA, USA). 

### 2.10. Liquid Chromatography–Mass Spectrometry (LC-MS) Analysis of the Ethanol Extract of O. linkii subsp. cyprius—Extract with the Highest Anti-Acanthamoeba Activity

Chromatographic separations were performed on a Dionex Ultimate 3000 RS Liquid Chromatography system on a Thermo Acclaim RSLC 120, C18 column (2.1 × 100 mm, 2.2 µm) with a binary gradient (A: water with 0.1% formic acid, B: acetonitrile with 0.1% formic acid) at 0.4 mL/min: 0–0.4 min: isocratic at 5% B, 0.4–9.9 min: linear from 5% B to 100% B, 9.9–15.0 min: isocratic at 100% B, 15.0–15.1 min: linear from 100% B to 5% B, 15.1–20.0 min: isocratic at 5% B. The injection volume was 2 µL. Eluted compounds were detected using a Dionex Ultimate DAD-3000 RS over a wavelength range of 200–400 nm and a Bruker Daltonics micrOTOF-QII time-of-flight mass spectrometer equipped with an Apollo electrospray ionization source in positive mode at 3 Hz over a mass range of *m*/*z* 50–1500 using the following instrument settings: nebulizer gas nitrogen, 2 bar; dry gas nitrogen, 9 L/min, 200 °C; capillary voltage 4500 V; end plate offset −500 V; transfer time 100 µs; prepulse storage 6 µs; collision gas nitrogen; collision energy 25 eV; collision RF 130 Vpp. Internal dataset calibration was performed for each analysis using the mass spectrum of a 10 mM solution of sodium formate in 50% isopropanol infused during LC re-equilibration using a divert valve equipped with a 20 µL sample loop. Data were analyzed using Bruker DataAnalysis 4.1 SP1.

### 2.11. Chemicals, Reagents, and Organisms

Quercetin, gallic acid, 2,5,7,8-tetramethylchroman carboxylic acid (Trolox), potassium persulfate, sodium chloride, p-iodonitrotetrazolium violet, chlorohexidine, ascorbic acid, trypan blue, 2,2-diphenyl-1-picrylhydrazyl (DPPH), (2,2′-azino-bis(3-ethylbenzothiazoline-6-sulfonic acid)) (ABTS), and aluminum chloride were obtained from Sigma-Aldrich St. Louis, MO, USA. Gentamicin sulfate, Mueller Hinton broth, Mueller Hinton agar, Ringer solution, and Folin–Ciocalteu reagent were purchased from Millipore Merck (Darmstadt, Germany). The brine shrimp eggs kit was purchased from a local pet shop. 

*Acanthamoeba castellanii* (ATCC 30010) and *Escherichia coli* (ATCC 25922) were purchased from the American Type Culture Collection (United States). Cell scrapers, 1.5 mL centrifuge tubes, and 96-well plates with black walls were from Santa Cruz (USA). Ethanol for extract preparation was of HPLC grade, ensuring its purity. The mobile phase solvents consisted of water, acetonitrile, and 0.1% formic acid, each of LC-MS purity grade. 

### 2.12. Data Analysis

The *A. castellanii* data were reported as the mean ± standard error of the mean (SEM) of triplicates subjected to one-way analysis of variance (ANOVA). The data were analyzed in GraphPad Prism for Windows, Version 10 (GraphPad Software, San Diego, CA, USA). The significant differences between the mean results of the various treatments and the control were determined by Dunnett’s multiple comparison test. A *p*-value < 0.05 was considered significant. The following screening assays were reported as the mean ± standard deviation of triplicates: radical scavenging assay, TPC, TFC, broth microdilution assay, and brine shrimp lethality assay, where the data were reported as the mean ± standard deviation (SD) for each assay. The linear regression coefficient (R^2^) for phenolic and flavonoid content with antioxidant activity was calculated by Microsoft Excel (USA).

## 3. Results

### 3.1. Radical Scavenging Activity, Total Phenolic Content, and Total Flavonoid Content of the Ethanolic Leaf Extracts

The ethanolic herbal extracts of *P. cham*. subsp. *cyprius* and *Q. alnifolia* demonstrated the highest antioxidant activities from the three endemic plants investigated, 0.629 ± 0.022 mg/mL and 0.155 ± 0.002 mg/mL, respectively (Table 2). *P. cham.* subsp. *cyprius* had the highest total phenol content of the study (64.58 ± 0.24 mg GAE/g), followed by *Q. alnifolia* (61.30 ± 1.80 mg GAE/g) and *O. linkii* subsp. cyprius (10.20 ± 0.24 mg GAE/g). Compounds in the *O. linkii* subsp. *cyprius* ethanolic leaf extract were less sensitive to the ABTS and DPPH radicals. The TFC values for *P. cham*. subsp. *cyprius* (195.32 ± 1.09 mg QE/g) and *O. linkii* subsp. *cyprius* (64.26 ± 1.48 mg QE/g) were high for demonstrating a moderate level of TPC. The observed discrepancy indicated a potential interference from other phytochemicals in this assay. LC-MS data were analyzed for the presence of flavonoids. Only a small amount of a single compound was found and tentatively identified as cynaroside.

### 3.2. Antibacterial Activity of O. linkii subsp. cyprius, P. cham. subsp. cyprius, and Q. alnifolia Ethanolic Leaf Extracts

Figure 2 illustrates that the ethanolic extract of *Q. alnifolia* demonstrated the most effective MIC against *S. aureus* (0.3125 mg/mL), *S. epidermis* (0.08 mg/mL), and *E. faecalis* (0.3125 mg/mL). It also demonstrated higher MICs against *B. subtilis* (1.25 mg/mL), *P. aeruginosa* (1.25 mg/mL), and *E. coli* (>2.5 mg/mL). The other ethanolic extracts of *O. linkii* subsp. *cyprius* and *P. cham.* subsp. *cyprius* were not adequate (MIC > 2.5 mg/mL) against the *S. aureus*, *S. epidermis*, *B. subtilis*, *P. aeruginosa*, and *E. coli*; however, they both generated a MIC of 0.625 mg/mL against *E. faecalis* (Figure 2).

### 3.3. Anti-Acanthamoeba Activity

Trypan blue stain (0.5%) was used to aid in the identification of viable and non-viable *Acanthamoeba* cells at 20× and 40× magnifications. After plant extract exposure, amoebae were easily observed undergoing morphological changes and absorption of trypan blue stain at 40× magnification (Figure 3). The highest IC_50_ values were observed at the higher doses and for increased exposure periods. *O*. *linkii* subsp. *cyprius* demonstrated the highest anti-*Acanthamoeba* activity after 48- and 72-h exposures as compared to control (no treatment), with respective to IC_50_ values of 19.21 mg/mL and 7.02 mg/mL (Figure 4A2,A3). *P. cham*. subsp. *cyprius* showed weak biological activity against *A. castellanii* at concentrations of 12.5 mg/mL and 25 mg/mL after 48- and 72-h exposures (Figure 4B2,B3). In this study, *Q. alnifolia* did not show biological activity against *A. castellanii* (Figure 4C1–C3). A dose–response effect on the % cell viability was observed when the *O. linkii* subsp. *cyprius* ethanolic extract concentration doubled after the 72-h exposure (Figure 5A3). No significant difference was observed between all plant treatment concentrations regardless of exposure periods at lower concentrations at the 24-h exposure period (Figure 5A1,B1,C1).

### 3.4. HepG2 Cell Viability Using Ethanolic Plant Extracts

The reduction of resazurin to resorufin was measured in a spectrofluorometer to record the % cell viability of human hepatocarcinoma (HepG2) cells after a 48-h exposure from individual ethanolic leaf extracts from *O. linkii* subsp. *cyprius*, *P. cham*. subsp. *cyprius,* and *Q. alnifolia*. The results were compared to the effects of the anti-*Acanthamoeba* drug miltefosine. *Q. alnifolia* (CC_50_ of 196.8 µg/mL) showed the highest cytotoxicity amongst the endemic plants screened in this study, followed by *P. cham*. subsp. *cyprius* (CC_50_ of 262.7 µg/mL). *O. linkii* subsp. *cyprius* did not appear to be acutely cytotoxic above 500 µg/mL within the 48-h exposure period (Figure 6A). The HepG2 cells tolerated the plant extracts better than miltefosine, which had a CC_50_ of 28.9 µg/mL (Figure 6B).

### 3.5. Extract Characterization of O. linkii subsp. cyprius

+ESI-LCMS was used for characterization. Untargeted peak detection using the Dissect-compounds-algorithm of DataAnalysis 4.1 (Bruker) yielded 95 chromatographic peaks, of which 14—accounting for 48% of the total peak area—were characterized. Results are shown in detail in Table 3 and Figure 7. The 14 chromatographic peaks contained adduct ions indicative of 16 chemical compounds, which were tentatively identified as the phenylethanoids, acteoside or isoacteoside, the flavone glucoside cynaroside, and 14 iridoid glycosides. Acteoside, cynaroside, and nine of the iridoids have been previously described as constituents of the related species *O. luteus* Steven, *O. vernus* (Bellardi) Dumort [28], *O. serotina* (Lam.) Dumort. [29], or *O. vulgaris* Moench [30,31]. The identity of the compounds was, in part, supported by +ESI reference spectra (mzCloud—Advanced Mass Spectral Database [32], but it was not possible to clearly distinguish between isomeric iridoids, so some of the assignments shown in Table 3 are interchangeable. The remaining five putative iridoid glycosides were identified as such by the neutral formulas generated from the accurate masses and isotope patterns of their clearly identified adduct ions. All spectra assigned to iridoid glycosides showed the signals of the sodiated molecule and at least one further adduct ion, usually the protonated, or the ammoniated molecule. In addition, all putative iridoids showed a neutral loss of C_6_H_10_O_5_ or C_6_H_12_O_6_, indicative of the loss of a hexosyl or hexose moiety, respectively. An unusual result was the surprising shortage of flavonoids or monocaffeoyl quinic acids, which is in contradiction to the TFC assay. 

For identification, the accurate molecular weight and the fragmentation patterns of the compounds in the peaks were cross-referenced with the mzCloud database for verification. Many of the identified compounds were iridoid glucosides with isomer groups (Figure 8; Photograph, Figure 9). Table 3 summarizes the accurate masses, formulas, and fragment ions found from targeted compounds that have been reported in the literature from other *Odontites* species that correspond to the compounds from the ethanolic leaf extract of *O. linkii* subsp. *cyprius* detected in this study. Phenylethanoids, acteoside or isoacteoside, and the flavone glycoside cynaroside were also detected. There was an unexpected shortage of phenolics, such as flavonoids and chlorogenic-acid-like compounds, in the ethanolic leaf extract (Figure 8). However, other compounds present in the extract could have formed interference with the aluminum chloride in the TFC assay and led to an overrated flavonoid content in the *O. linkii* subsp. *cyprius* ethanolic leaf extract. 

## 4. Discussion

### 4.1. Need for New Pharmaceutical Discoveries for the Treatment of Acanthamoeba spp. Infections

*Acanthamoeba* spp. are opportunistic protists known to cause severe keratitis among contact lens wearers and can trigger cutaneous lesions, especially among immunocompromised individuals [1,2,3]. They can also cause fatal granulomatous amoebic encephalitis (GAE) in humans with compromised immune systems [8,9]. 

*Acanthamoeba* species are commonly found in soil, water, and a variety of environments, including medical and industrial settings [7,13,33]. They can withstand a range of osmolarities and survive in distilled water, tissue culture media, and bodily fluids [3,5,34]. Their ability to endure harsh conditions is attributed to their survival mechanisms, particularly the double-wall cyst stage, which enables them to withstand hostile environments and resist treatments [5,16]. This can result in repeated patient infections and potentially life-threatening situations [16]. Cysts of *Acanthamoeba* spp. have been found in chlorinated swimming pools, domestic tap water, soil, air, sewage, seawater, bottled water, hospitals, air-conditioning units, and contact lens cases [4,6,12,14]. They have also been isolated from nasal passages, skin, intestines, plants, and other vertebrates [4]. The lack of effective chemotherapies and often late diagnosis makes *Acanthamoeba* spp. infections particularly challenging to treat, underscoring the urgency to find better treatment methods [15]. Many plant-based natural products highlight the potential for the discovery of new pharmaceuticals or insight into mechanisms for the treatment of pathogenic protozoan infections.

### 4.2. Antioxidant Capacity of the Ethanol Leaf Extracts

The ethanolic leaf extract from *P. cham*. subsp. *cyprius* demonstrated the highest TFC (195 ± 1.09 mg QE/g), which correlated relatively well with the TPC (64.58 mg GAE/g), DPPH (0.629 ± 0.022 mg/mL), and ABTS (0.596 ± 0.262 mg/mL) assays. The ethanolic leaf extract from *Q. alnifolia* showed the highest radical scavenging activities (DPPH IC_50_ = 0.155 ± 0.002 mg/mL; ABTS IC_50_ = 0.164 ± 0.017 mg/mL) that correlated with the TPC (61.30 ± 1.80 mg GAE/g) and TFC (51.34 ± 1.09 QE/g) results. Screening the ethanolic leaf extract of *O. linkii* subsp. *cyprius* for DPPH (IC_50_ = 9.889 ± 2.545 mg/mL) and ABTS (IC_50_ = 3.235 ± 0.655 mg/mL) radical scavenging activities poorly correlated with the total phenolic content (10.20 ± 0.24 mg GAE/g) and TFC (64.26 ± 1.48 mg QE/g). The *O. linkii* subsp. *cyprius* ethanolic extract appeared only to weakly scavenge radicals in both assays. These results did not correlate with the relatively high total flavonoid content observed in the TFC aluminum chloride assay. Aluminum chloride (AlCl_3_) can form complexes that involve the 5-OH (or 3-OH) and 4-carbonyl groups of flavonoids, especially if the molecules have vicinal phenolic hydroxyl (OH) groups or an OH-group in *peri*-position to a carbonyl group. It is possible that other phytochemicals, including pigments, were present in the *O. linkii* subsp. *cyprius* ethanolic extract simulated this situation and led to an overrated response in the TFC assay [35,36]. Moreover, the LC-MS analysis did not reveal a high presence of flavonoid compounds in the *O. linkii* subsp. *cyprius* extract at 340 nm. The *O. linkii* subsp. *cyprius* leaves were black after drying, indicating the breakdown of iridoid compounds. There was a possibility that the iridoid breakdown products interfered with the assay; however, due to the usually high level of perceived total flavonoid content in the *O. linkii* subsp. *cyprius* extract from the original TFC assay, this resulted in further screening for biological activity with reference pathogenic microorganisms and the other endemic ethanolic leaf extracts used in this study.

### 4.3. Broth Microdilution Antibacterial-Susceptibility Test of O. linkii subsp. cyprius, P. cham. subsp. cyprius, and Q. alnifolia 

According to the late 16th-century manuscript called the ‘Iatrosophikon’ (medical wisdom) of the Makhairas monastery in Cyprus, *Q. alnifolia* was the only endemic plant used in this study that was mentioned in this medical manuscript. The leaf galls of the golden oak (*Q. alnifolia*) were traditionally combined with other herbal ingredients to treat skin wounds [37], which correlates with ethanolic leaf extract antibacterial properties against the Gram-positive pathogenic bacteria used in this study (*E. faecalis*, *S. aureus*, and *S. epidermis*; MIC < 0.3125 mg/mL, respectively). Extracts from other *Quercus* species have demonstrated antibacterial properties (see reviews by [38,39,40]). The other endemic plants, *O. linkii* subsp. *cyprius* and *P. cham*. subsp. *cyprius*, were not mentioned in the ‘Iatrosophikon’ herbal formulations, perhaps due to their ineffectiveness against bacteria that typically colonize the skin. However, these plant extracts did show moderate effectiveness against *E. faecalis* (MIC = 0.625 mg/mL, respectively) and ineffectiveness against the other pathogenic bacteria used in this screening study (MIC > 2.5 mg/mL). In general, many researchers have rated antibacterial activity from plant extracts with MIC values of ≤0.1 mg/mL as significant and <0.625 mg/mL as moderate [41]. After the ethanolic leaf extracts demonstrated antibacterial properties, they were further screened for anti-*Acanthamoeba* activity and LC-MS analysis for known bioactive compounds. 

### 4.4. Anti-Acanthamoeba % Cell Viability Assay

A dose–response relationship was only observed for *O. linkii* subsp. *cyprius* and *P. cham*. subsp. *cyprius* ethanolic leaf extracts at concentrations ranging from 3.125 to 25 mg/mL against *A. castellanii* trophozoites after a 24-, 48- and 72-h incubation period. At the highest dose of 25 mg/mL, *O. linkii* subsp. *cyprius* demonstrated the strongest anti-*Acanthamoeba* activity after 72 h (>78% inhibition), followed by *P. cham*. subsp. *cyprius* (>60% inhibition). The *A. castellanii* trophozoite were not sensitive to the *Q. alnifolia* ethanolic leaf extract (>60% inhibition), which signifies a typical resistance response to plant extracts. In a related study using another *Quercus* species, Sawangjaroen et al. (2004) [42] reported a very low cure rate of 26% (4/15) after a 5-day treatment using a 1000 µg dose from the methanol extract of *Quercus infectoria* (Oliv.) nut galls in mice that were infected with another free-living amoeba, *Entamoeba histolytica*, which had been injected directly into the caecum of peritoneal cavity during their study. In our study, *O. linkii* subsp. *cyprius* demonstrated the best in vitro activity against the *A. castellanii* reference strain with an IC_50_ value of 7.02 mg/mL after a 72-h exposure. This is the first report of anti-*Acanthamoeba* activity from ethanolic leaf extracts from *Odontites linkii* subsp. *cyprius* and *P. cham*. subsp. *cyprius*. Other studies have reported in vitro anti-*Acanthamoeba* activity from plant extracts [27,43,44,45,46]. Combining these plant extracts with standard pharmacological anti-amoebic treatments may improve the outcome, especially in treatment-resistant cases. 

### 4.5. Safety and Efficacy of the Ethanolic Leaf Extracts

According to the results of the HepG2 cell viability assay, the cells tolerated the highest concentration of 500 µg/mL *O. linkii* subsp. *cyprius* ethanolic leaf extract (IC_50_ = >500 µg/mL) after a 48-h exposure in this study. However, the cells showed more chemosensitivity to the ethanolic leaf extracts from *P. cham*. subsp. *cyprius* (IC_50_ = 262.7 µg/mL), *Q. alnifolia* (IC_50_ = 196.8 µg/mL), and the reference drug, miltefosine (IC_50_ = 28.9 µg/mL). Oral miltefosine has recently been suggested as an adjunct treatment for refractory *Acanthamoeba* keratitis [17,18,47]. Additional toxicity investigation should be continued to establish the safety and efficacy of the ethanolic leaf extract of *O. linkii* subsp. *cyprius* as a supporting treatment for *Acanthamoeba* infections. 

### 4.6. LC-MS Screening for Known Bioactive Compounds in O. linkii subsp. cyprius

Preliminary LC-MS analysis reveals several iridoids compounds with many isomer groups that can be found in the identified and unidentified peaks. After interpreting the fragmentation pattern from the collected mass spectra of the *O. linkii* subsp. *cyprius* ethanolic leaf extract, sixteen compounds accounting for nearly 50% of the total peak area and were characterized or tentatively identified. The isomeric iridoid glycosides most likely differ in the position or orientation of hydroxy groups. The isomers could not be distinguished, which made concise MS assignments within an isomer group impossible in this preliminary study.

LC-MS chromatograms of *O. linkii* subsp. *cyprius* ethanolic leaf extract revealed peaks of phenylethanoids, acteoside, or isoacteoside, and the flavone glycoside cynaroside. Notably, the extract lacked phenolics, such as flavonoids and chlorogenic-acid-like compounds, which were a surprising find. This phenomenon was consistent with results from radical scavenging assays (DPPH IC_50_ = 9.889 ± 2.545 mg/mL; ABST IC_50_ = 3.235 ± 0.655 mg/mL) and total phenolic content assay (10.20 ± 0.24 mg GAE/g) but not the total flavonoid content assay (64.26 ± 1.48 mg QE/g). The presence of other phytochemicals in the ethanolic extract likely led to an overrated result in the assay when they reacted with the AlCl_3_. However, the synergistic effect of all compounds in the extract, including the iridoid derivatives, the phenylethanoids, acteoside, and the flavone glycoside cynaroside, could have contributed to the *in vitro* anti-*Acanthamoeba* activity (IC_50_ of 7.02 mg/mL after 72 h) and antibacterial activity against *E. faecalis* (MIC 0.625 mg/mL). This LC-MS analysis adds to the phytochemical data of *O. linkii* subsp. *cyprius* in the genus *Odontites* and the family of *Orobanchaceae*.

Liu et al. (2023) [48] isolated several compounds from *O. vulgaris* Moench with moderate anti-acetylcholinesterase (anti-AChE) activity for the exploration of new Alzheimer’s disease treatments. Ji et al. (2021) [30] purified and tested *O. vulgaris* Moench for anti-inflammatory activity, where verbascoside (synonymous with acteoside), isoacteoside, melampyroside, and tricin were identified from the ethyl acetate and n-butanol fractions. Verbascoside, isoacteoside, and phenylethanolglycosides have been considered sources for pharmaceutical development for the treatment of rheumatoid arthritis after they were shown to have significant anti-inflammatory activity [30]. Verbascoside has also been investigated for anticancer properties [49,50]. Iridoid compounds, such as melampyroside, have also been studied for their anti-inflammatory activity [51]. Our preliminary LC-MS analysis also showed many of these compounds in the *O. linkii* subsp. *cyprius* ethanolic leaf extract.

The isolation of bioactive compounds from *O. linkii* subsp. *cyprius* and *P. cham.* subsp. *cyprius* presents an exciting opportunity for testing and potential drug discovery. These iridoid compounds, with their promising properties, could pave the way for the development of new antimicrobial and antioxidant agents. However, it is crucial to emphasize that further studies are necessary to confirm the antibacterial and anti-*Acanthamoeba* results of this preliminary study. This research underscores the continuous development of natural products into novel and effective pharmaceuticals [52,53].

## 5. Conclusions

This is the first report of the *Odontites linkii* subsp. *cyprius* and *Ptilostemon chamaepeuce* subsp. *cyprius* with anti-*Acanthamoeba* activity. Combining these plant extracts with standard pharmacological anti-amoebic treatments may improve patient outcomes, especially in treatment-resistant cases. Phytochemical analytical studies and bioassay-guided fractionation are recommended to identify the biologically active compounds against these pathogens. Preliminary LC-MS analysis of *Odontites linkii* subsp. *cyprius* has revealed potent iridoid compounds that are associated with anti-inflammatory properties. Additional investigation into the toxicity of these plants needs to be performed to establish application for pharmaceutical-leads studies.

## Figures and Tables

**Figure 1 microorganisms-12-02303-f001:**
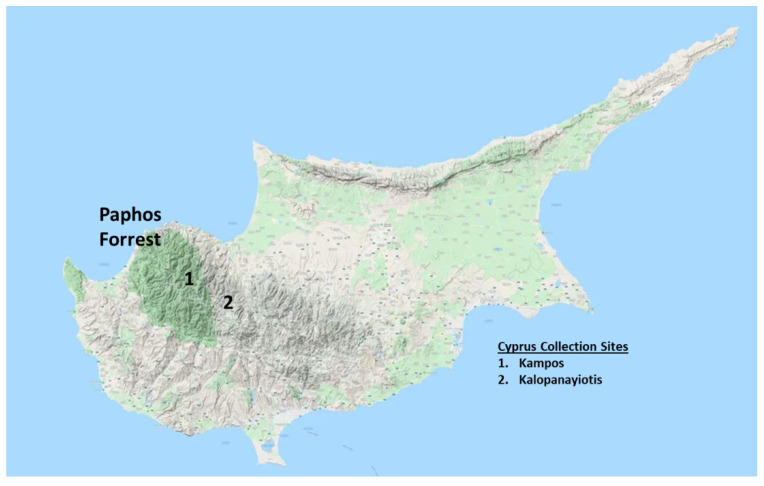
Plant collection sites in Cyprus.

**Figure 2 microorganisms-12-02303-f002:**
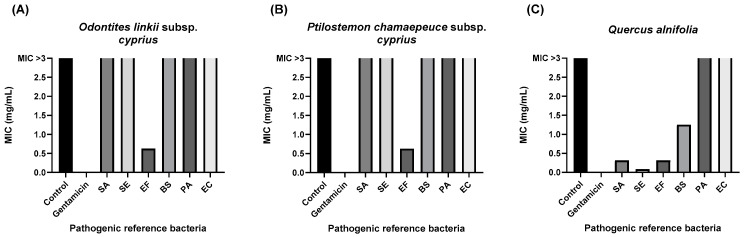
Antibacterial activity of ethanolic leaf extracts at concentrations 0–2.5 mg/mL. (**A**) *Odontites linkii* subsp. *cyprius*. (**B**) *Ptilostemon chamaepeuce* subsp. *cyprius*. (**C**) *Quercus alnifolia*. Minimum inhibitory concentration (MIC) in mg/mL. Gram-positive bacteria: SA: *Staphylococcus aureus* (ATCC 6538), SE: *Staphylococcus epidermis* (ATCC 12228), BS: *Bacillus subtilis* (ATCC 6633), EF: *Enterococcus faecalis* (ATCC 29212). Gram-negative bacteria: PA: *Pseudomonas aeruginosa* (ATCC 27853), EC: *Escherichia coli* (ATCC 25922). Gentamicin was used as the positive control from 0.02–2.5 mg/mL. All bacteria were sensitive to gentamicin less than 0.02 mg/mL.

**Figure 3 microorganisms-12-02303-f003:**
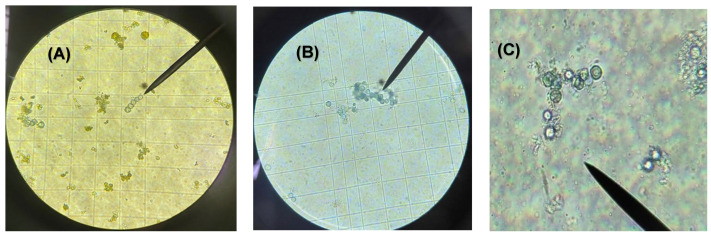
*Acanthamoeba castellanii* (ATCC 30010) trophozoite chemosensitivity and viability towards ethanolic extract of *Odontites linkii* subsp. *cyprius* after a 72-h exposure at 6.25 mg/mL. Cells that absorbed the trypan blue stain were damaged or dead. (**A**) *A. castellanii* trophozoite exposure to *O. linkii* subsp. *cyprius* without trypan blue stain at 20× magnification. (**B**) *A. castellanii* trophozoite exposure to *O. linkii* subsp. *cyprius* with 0.5% trypan blue stain at 20× magnification. (**C**) *A. castellanii* trophocyte exposure to *O. linkii* subsp. *cyprius* with 0.5% trypan blue stain at 40× magnification.

**Figure 4 microorganisms-12-02303-f004:**
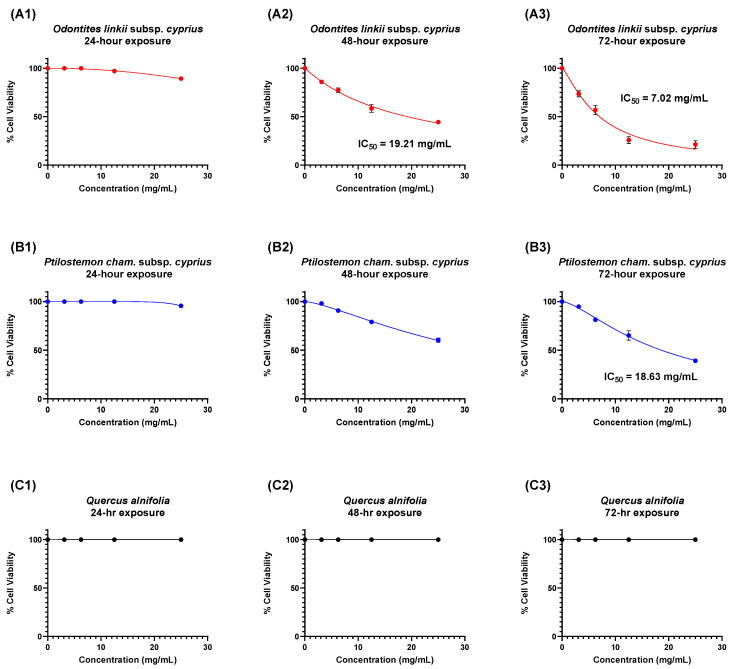
Dose–response curves of anti-*Acanthamoeba* trophozoite % cell viability assay with ethanolic leaf extracts at various time points. (**A**) *Odontites linkii* subsp. *cyprius*. (**B**) *Ptilostemon chamaepeuce* subsp. *cyprius*. (**C**) *Quercus alnifolia*. (**1**) 24-h exposure effect. (**2**) 48-h exposure effect. (**3**) 72-h exposure effect. Results are shown as mean ± standard error of mean (*n* = 3). *Acanthamoeba castellanii* (ATCC 30010). IC_50_ = inhibitory concentration at 50%. % cell viability = (sample − control)/(control) × 100.

**Figure 5 microorganisms-12-02303-f005:**
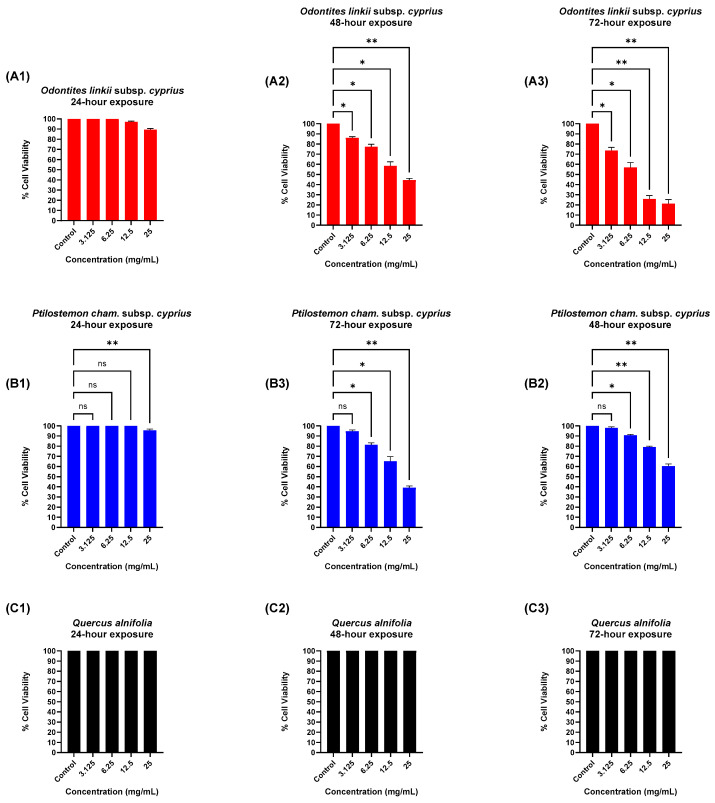
*Acanthamoeba castellanii* (ATCC 30010) trophozoite % cell viability after exposure to ethanolic leaf extracts at various time points. (**A**) *Odontites linkii* subsp. *cyprius*. (**B**) *Ptilostemon chamaepeuce* subsp. *cyprius*. (**C**) *Quercus alnifolia*. (**1**) 24-h exposure effect. (**2**) 48-h exposure effect. (**3**) 72-h exposure effect. Data expressed as mean ± standard error of mean (*n* = 3). One-way ANOVA with Dunnett’s multiple comparison test. *p*-value ≤ 0.05. Mean values with an asterisk in the column are statistically significant. ns: non-significant compared to control. * Mild significance. ** Moderate significance.

**Figure 6 microorganisms-12-02303-f006:**
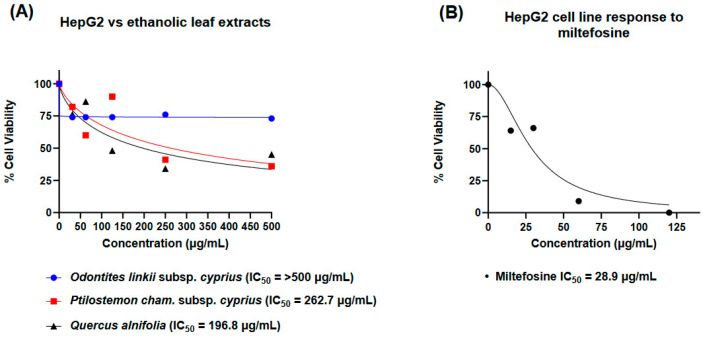
Human hepatocarcinoma cell line (HepG2) dose–response to ethanolic extract concentrations 0–500 µg/mL after 48-h exposure. (**A**) HepG2 response to selected endemic plants from Cyprus: *Odontites linkii* subsp. *cyprius*, *Ptilostemon chamaepeuce* subsp. *cyprius*, and *Quercus alnifolia*. (**B**) HepG2 response to miltefosine (anti-*Acanthamoeba* reference drug for *Acanthamoeba* keratitis) at 0–120 µg/mL concentrations. % Cell viability expressed as (control − treatment)/(control) × 100. The assay was performed in triplicate for each plant extract and control.

**Figure 7 microorganisms-12-02303-f007:**
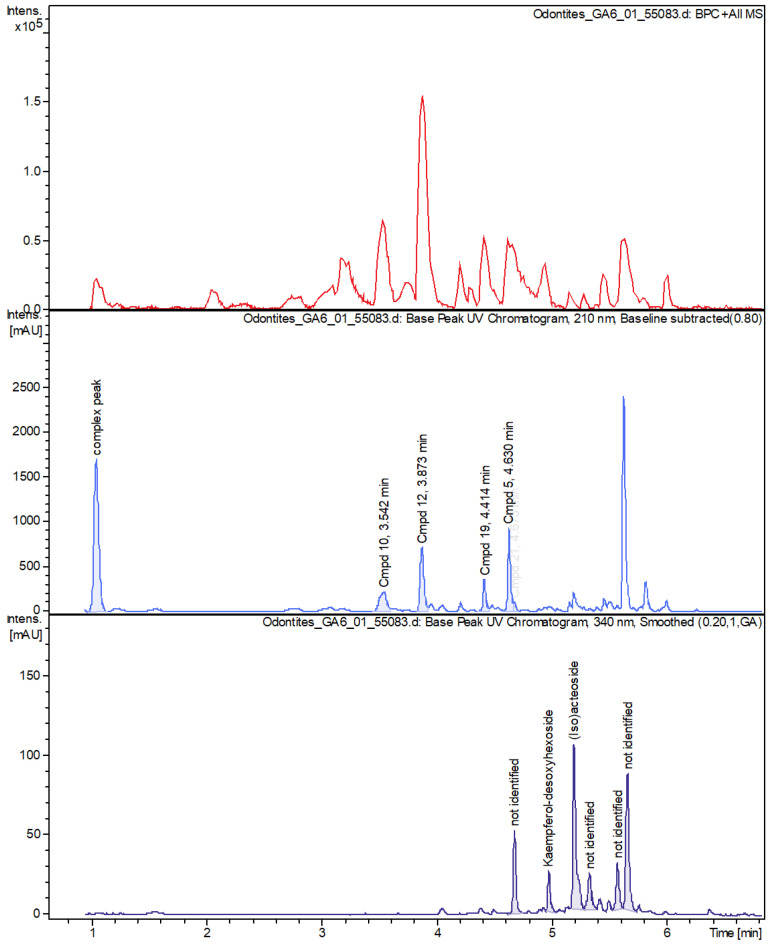
*Odontites linkii* subsp. *cyprius* ethanolic leaf extract chromatographs. Base peak chromatogram (**upper panel**). UV at 210 nm (**middle panel**). UV at 340 nm (**lower panel**). The complex peak consists of several chemical compounds, including iridoid glycosides.

**Figure 8 microorganisms-12-02303-f008:**
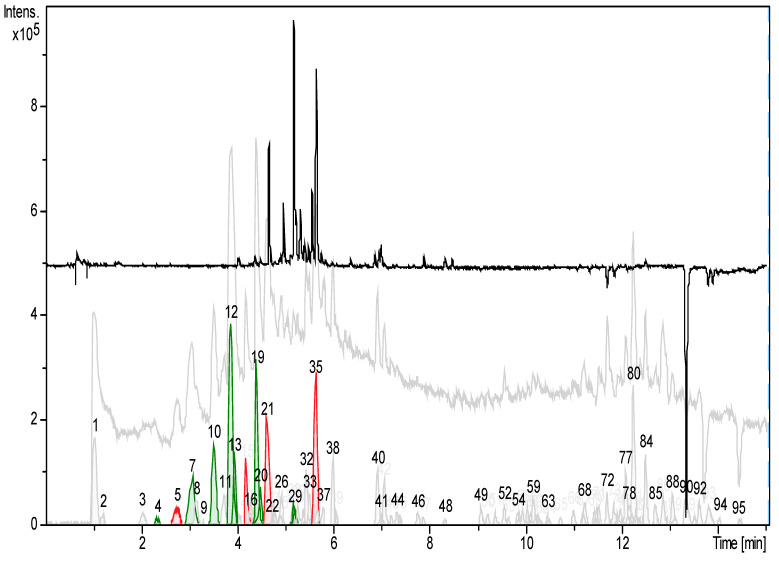
+ESI-LCMS analysis of ethanolic extract of *O. linkii* subsp. *cyprius*. Peaks assigned based on known compounds of *Odontites* spp. are colored green, and red-colored peaks have been amended. The grey-colored continuous plot is the background-subtracted total ion chromatogram (TIC). The black-colored plot is a UV chromatogram at 340 nm; neither UV- nor MS-spectra indicates the presence of flavonoids other than cynaroside (peak 29).

**Figure 9 microorganisms-12-02303-f009:**
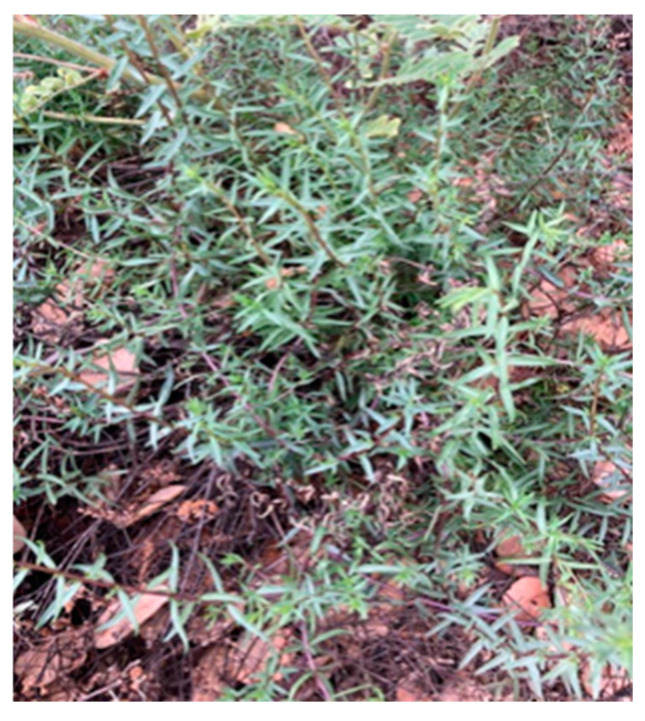
Photograph of *Odontites linkii* subsp. *cyprius* with iridoid compounds that showed in vitro anti-*Acanthamoeba* activity after 72-h exposure.

**Table 1 microorganisms-12-02303-t001:** Endemic plants collected from Cyprus.

Plant Name	Family	Status	Collection Site	Voucher No.	Month/Year
*Odontites linkii* subsp. *cyprius* (Boiss.) Bolliger	*Orobanchaceae*	Endemic	Kampos	MS20	June 2022
*Ptilostemon chamaepeuce* subsp. *cyprius* (Greuter) Chrtek & B. Slavík	*Asteraceae*	Endemic	Kalopanayiotis, roadside	MS21	June 2022
*Quercus alnifolia* Poech	*Fagaceae*	Endemic	Vavatsinia	MS13	June 2022

**Table 2 microorganisms-12-02303-t002:** Radical scavenging assays (DPPH and ABST), total phenolic content (TPC), and total flavonoid content (TFC) of ethanolic leaf extracts.

Plant Name	Extraction Yield (%)	DPPH IC_50_ (mg/mL)	ABTS IC_50_ (mg/mL)	* TPC (mg GAE/g)	** TFC (mg QE/g)
*O. linkii* subsp. *cyprius*	17.3	9.889 ± 2.545	3.235 ± 0.655	10.20 ± 0.24	64.26 ± 1.48
*P. cham.* subsp. *cyprius*	15.2	0.629 ± 0.022	0.596 ± 0.262	64.58 ± 0.24	195.32 ± 1.09
*Q. alnifolia*	30.3	0.155 ± 0.002	0.164 ± 0.017	61.30 ± 1.80	51.34 ± 1.09
Ascorbic acid	-	0.036 ± 0.000	-	-	-
Trolox	-	0.047 ± 0.001	0.076 ± 0.001	-	-
Quercetin	-	-	0.217 ± 0.005	-	-

For IC_50_, *n* = 3, mean ± std value. * Gallic Acid Standard Curve R^2^ = 0.9971. ** Quercetin Standard Curve R^2^ = 0.9977. TPC (total phenolic content). TFC (total flavonoid content).

**Table 3 microorganisms-12-02303-t003:** Compounds tentatively identified in the ethanolic extract of *O. linkii* subsp. *cyprius* by +ESI-LC-MS.

No	*t*_R_/min	*m*/*z*	Ion	Formula	Err/mDa	mΣ	Assignment
4a	2.38	399.1265 215.0908	[M+Na]^+^	C_16_H_24_O_10_	0.3	46.7	Adoxosidic acid *^a^*
4b	2.38	405.1396 243.0877	[M+H]^+^	C_17_H_24_O_11_	−0.4	7.3	Methyloleoside *^a^*
5	2.76	377.1443 215.0919	[M+H]^+^	C_16_H_24_O_10_	0.1	17.9	Unknown C10-Iridoidglucoside
6	2.80	394.1714 215.0919	[M+NH_4_]^+^	C_16_H_24_O_10_	0.6	36.6	Unknown C10-Iridoidglucoside
7a	3.08	377.1456 215.0919	[M+H]^+^	C_16_H_24_O_10_	1.4	18.5	8-Epi-loganic acid *^a,b^*
7b	3.08	407.1562 227.0932	[M+H]^+^	C_17_H_26_O_11_	−1.4	21	Caryoptoside *^a^*
10	3.53	389.1441 227.0924	[M+H]^+^	C_17_H_24_O_10_	−0.1	31.5	Gardoside methyl ester *^a,b^*
12	3.87	407.1546 227.0920	[M+H]^+^	C_17_H_26_O_11_	0.1	21.9	Shanzhiside methyl ester *^a^*
13	3.95	331.1361 151.0764	[M+H]^+^	C_15_H_22_O_8_	−2.7	35.3	Bartsioside *^a,b^*
15	4.21	429.1336 227.0915	[M+Na]^+^	C_17_H_26_O_11_	3.1	31.3	8-O-Acetylharpagide
19	4.41	391.1568 229.1082	[M+H]^+^	C_17_H_26_O_10_	−3.1	22.4	8-Epiloganine *^a,b^*
20	4.5	391.1567 229.1071	[M+H]^+^	C_17_H_26_O_10_	3.1	–	Mussaenoside *^a^*
21	4.63	408.1874 211.0962	[M+NH_4_]^+^	C_17_H_26_O_10_	−1	37.9	Unkown C10-Iridoidglucoside
28	5.19	625.2168	[M+H]^+^	C_29_H_36_O_15_	−4.1	36.2	(Iso)acteoside *^a^*
29	5.21	449.1064 287.0535	[M+H]^+^	C_21_H_20_O_11_	−1.4	53.1	Cynaroside *^a,b^*
35	5.64	468.1865 271.0953	[M+NH_4_]^+^	C_22_H_26_O_10_	−0.1	29.0	Melampyroside

*^a^* Known constituent of the genus *Odontites*. *^b^* Identity supported by *m*/*z*-cloud database. Assignments for compounds with the same formula are interchangeable. The formula was determined from each compound’s most abundant adduct ion. The second *m*/*z*-values given for iridoid glycosides and cynaroside represent the respective protonated aglycon fragment, with a neutral loss of 162 u or 180 u from the protonated molecule. Compound numbers amended by *^a^* and *^b^* were found in the same chromatographic peak.

## Data Availability

The data are contained within the article; further inquiries can be directed toward the corresponding authors.

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
