# Peer review of "Odontites linkii subsp. cyprius Ethanolic Extract Indicated In Vitro Anti-Acanthamoeba Effect"

_microorganisms, 2024, doi:10.3390/microorganisms12112303_

Round 1

Reviewer 1 Report

Comments and Suggestions for Authors

Amoebic keratitis is a rare infection of the cornea (the transparent layer in front of the iris and pupil of the eye) caused by species of Acanthamoeba, free-living amoebas. It usually occurs in people who wear contact lenses. Amoebic keratitis causes painful ulcerations on the cornea and vision is often affected.

Facultative or opportunistic parasitic amoebae, or better known as free-living amoebae, can cause meningoencephalitis, granulomatous encephalitis and corneal ulcers. It is quite likely that human cases have been occurring for a long time, but we can think that we are dealing with protozoa that have transitioned to parasitic life, which increases the need for research to better understand these new parasites, especially their pathogenesis and epidemiology. These amoebas are: Naegleria fowleri, which is pathogenic; Acanthamoeba sp, which belongs to the Hartmanellidae family with eight species of medical interest; Balamuthia mandrilaris and Leptomyxa, which belong to the Leptomyxidae family.

Facultative or opportunistic parasitic amoebas, or better known as free-living amoebas, can cause meningoencephalitis, granulomatous encephalitis and corneal ulcers. It is very likely that human cases have been occurring for a long time, but we can think that we are dealing with protozoa transitioning to parasitic life, which increases the need for research to better understand these new parasites, especially their pathogenesis and epidemiology. These amoebas are: Naegleria fowleri, which is pathogenic; Acanthamoeba sp, which belongs to the Hartmanellidae family with eight species of medical interest; Balamuthia mandrilaris and Leptomyxa, which belong to the Leptomyxidae family.

Amoebas (Acanthamoeba sp) only invade previously damaged corneas. If there is extensive abrasion, keratitis develops rapidly, with corneal ulceration, severe pain and variable loss of vision. In people who wear contact lenses, small lesions caused by the incorrect use of this visual corrective agent can allow the penetration of amoebas from contaminants in saline solutions prepared at home, for example.

These amoebas are found in soil and in the water of lakes and rivers. The trophozoite forms are active and feed on bacteria; the cysts are found in dry soil or dust. Excystment occurs when the cysts enter a humid environment, mainly in the presence of other bacteria. The flagellated forms are very active in water and, when they come into contact with the nasal mucosa, they transform into active trophozoites, which reach the brain via the neuroolfactory epithelium, where they spread through the bloodstream.

Meningoencephalitis in humans can present in acute and chronic forms. Acute cases are caused by Naegleria fowleri, which causes hemorrhagic and necrotic lesions affecting the base of the frontal lobe and brain. It is commonly associated with young swimmers. Chronic cases have been associated with Acanthamoeba sp, which causes granulomatous amoebic encephalitis in weakened and immunocompromised patients. The only way to prevent amoebic encephalitis is to avoid swimming in lakes, pools and rivers that are susceptible to contamination.

Treatment is problematic because there is no known specific medication that is effective, although several drugs have been tested. The drugs with some effect and used in patients were amphotericin B, miconazole and rifampicin, administered intravenously and intrathecally. In cases of keratitis, treatment is also difficult, and the following are used topically: propamidine isethionate, polyhexamethylene biguanide and neomycin, and orally: ketoconazole or itraconazole.

This manuscript submitted by Chad Schou et al. describes for the first time the use of Odontites linkii subsp. cyprius and Ptilostemon chamaepeuce subsp. cyprius with anti-Acanthamoeba activity. The authors demonstrated that the combination of these plant extracts with standard antiamebic pharmacological treatments improves patient outcomes, especially in treatment-resistant cases. Phytochemical analytical studies and bioassay-guided fractionation are recommended to identify the biologically active compounds against these pathogens. Preliminary LC-MS analysis of Odontites linkii subsp. cyprius revealed potent iridoid compounds that are associated with anti-inflammatory properties. Further investigations into the toxicity of these plants need to be carried out to establish their application for pharmaceutical leadership studies.

The manuscript is interesting and extremely relevant to the field of study.

Author Response

Dear Reviewer,

Thank you for providing your valuable feedback and insight about our manuscript.  

We are made all the necessary corrections according to your comments, including the manuscript formatting in the reference section according to the journal's guidelines.  

We have attached the manuscript with track changes.  We have also attached the clean version.

Thanks again for providing your valuable feedback.

Best Regards.

Reviewer 2 Report

Comments and Suggestions for Authors

MAJOR POINTS

1.     The abstract

·       correct the abbreviation for

·       “…, total phenolic content (TFC)…” (page 1, line 15) into TPC

·       hours (page 1, line 21) – hr into hrs

2.     Keywords

·       increase the number of keywords

·       for example, leaf extracts

·       better define keywords

·       anti-Acanthamoeba effect

·       in vitro - in Italic

3.     In the introduction – the number of references should be increased

·       Sentence „ Acanthamoeba species (spp.) are opportunistic protists and known to cause severe keratitis among contact lens wearers and can trigger cutaneous lesions, especially among immunocompromised individuals. “ - (page 1, lines 38-41) need reference!

·       Sentence „ Infections from free-living amoebas are not considered neglected tropical diseases but can become a public health concern if environmental circumstances encourage reproductive conditions. Acanthamoeba spp. infections are the most common cause of Acanthamoeba keratitis (AK). “- (page 1, lines 43-46) need reference!

·       “…a dormant, stress-resistant cyst, which has allowed it to be isolated from chlorinated swimming pools, domestic tap water, soil, air, sewage, seawater, bottled water, hospitals, air-conditioning units, and contact lens cases.” need to have reference!

·       Describe the aim of the study in more sentences.

1.     Discussion

·       Rephrase repetitive sentences from the introduction!

·       “Acanthamoeba spp. are opportunistic protists known to cause severe keratitis among contact lens wearers…” – page 15, lines 480-481

·       “Acanthamoeba spp. cysts have been recovered chlorinated swimming pools, domestic tap water, soil, air, sewage, seawater, bottled water, hospitals, air-conditioning units, and contact lens cases. Isolates have also been recovered from nasal cavities, skin, intestines, as well as plants and other vertebrates [1] - page 16, lines 487-490

·       Acanthamoeba spp. have emerged as a human pathogen. - page 16, lines 492-493

2.     The titles

·       Rephrase this title better

·       4.2. Antioxidants

·       4.6. LC-MS

 MINOR POINTS

1.     Coordinate the entire paper according to the Instructions for the authors of this journal.

·       Correct - “(See reviews by Burlacu et al. (2020) [26]; 533 ÅžöhretoÄŸlu and Renda (2020) [27]; Taib et al. (2020 [28]).” – page 16, lines 533-534

·       Reference correction

                                                    i.     “[31-33], [17], [34]” – page17, line 562.

                                                  ii.     “[7], [8], [35]” - page17, line 572.

·       Figures / Tables

                                                    i.     The position of figures should be either above their first explanation in the text or after their first mention.

                                                  ii.     Figures - 4, 5 and 6 should be merged in one (A, B and C) - different colors for different leaves

                                                iii.     Figures - 7, 8 and 9 should be merged in one (A, B and C) - different colors for different leaves

2.     Latin abbreviation „et al.“ should be written in Italic – throughout the paper

3.     Enclose software specification in parentheses (manufacturer, city, country) – page 6, line 265; page 7, line 292

4.     Correct - write the numbers in the chemical formula in the index

·       page 12, line 437; Table 3.; page 16, line 510.

5.     Technically correct the entire paper

·       the space between the point and the first word of another sentence – throughout the paper - page 2, lines 51 and 66; page 4, lines 129 and 170

·       the space between the number and the measurement unit –

                                                    i.     “…340nm” - page 16, lines 516;

6.     The words in vitro should be written in Italic type since they are Latin words – page 5, line 202

Author Response

Dear Reviewer,

Thank you for providing your valuable feedback and insight about our manuscript.  We have addressed all your comments.  

We have attached the manuscript with track changes that address the typos and guideline formatting issues.  We have also attached the clean version for your review.  

Major Points

  1. Typos corrected.
  2. Keywords have been increased.
  3.  Supporting references have been added in the introduction, including the explanation of the aim of the study.
  4.  Sentences have been rephrased in the discussion section, including supporting references.
  5. Titles in 4.2 and 4.6 paragraphs have been rephrased.

Minor Points

  1. Paper has been coordinated with Instructions for the authors of the journal, including the reference and combination of figures.
  2. Italics were used for "et al." when mentioned in the manuscript.
  3. GraphPad software information included (manufacturer, city, country).
  4. We added the chemical formulas to the index.
  5. The typo spaces between words have been corrected throughout the manuscript.
  6. Italics script was used for in vitro on page 5, line 202.

Thank you again for providing your feedback.
